# High Catalytic Efficiency of a Layered Coordination Polymer to Remove Simultaneous Sulfur and Nitrogen Compounds from Fuels

**Fátima Mirante** [1] , **Ricardo F. Mendes** [2] , **Filipe A. Almeida Paz** [2,*] and **Salete S. Balula** [1,*]

1    REQUIMTE/LAQV & Department of Chemistry and Biochemistry, Faculty of Sciences, University of Porto, 4169-007 Porto, Portugal; fatimaisabelmirante@gmail.com
2    CICECO-Aveiro Institute of Materials, Department of Chemistry, University of Aveiro, Campus Universitário de Santiago, 3810-193 Aveiro, Portugal; rfmendes@ua.pt
*    Correspondence: filipe.paz@ua.pt (F.A.A.P.); sbalula@fc.up.pt (S.S.B.)

**Abstract:** An ionic lamellar coordination polymer based on a flexible triphosphonic acid linker, [Gd(H$_4$nmp)(H$_2$O)$_2$]Cl$_2$ H$_2$O (1) (H$_6$nmp stands for nitrilo(trimethylphosphonic) acid), presents high efficiency to remove sulfur and nitrogen pollutant compounds from model diesel. Its oxidative catalytic performance was investigated using single sulfur (1-BT, DBT, 4-MDBT and 4,6-DMDBT, 2350 ppm of S) and nitrogen (indole and quinolone, 400 ppm of N) model diesels and further, using multicomponent S/N model diesel. Different methodologies of preparation followed (microwave, one-pot, hydrothermal) originated small morphological differences that did not influenced the catalytic performance of catalyst. Complete desulfurization and denitrogenation were achieved after 2 h using single model diesels, an ionic liquid as extraction solvent ([BMIM]PF$_6$) and H$_2$O$_2$ as oxidant. Simultaneous desulfurization and denitrogenation processes revealed that the nitrogen compounds are more easily removed from the diesel phase to the [BMIM]PF$_6$ phase and consequently, faster oxidized than the sulfur compounds. The lamellar catalyst showed a high recycle capacity for desulfurization. The reusability of the diesel/H$_2$O$_2$/[BMIM]PF$_6$ system catalyzed by lamellar catalyst was more efficient for denitrogenation than for desulfurization process using a multicomponent model diesel. This behavior is not associated with the catalyst performance but it is mainly due to the saturation of S/N compounds in the extraction phase.

**Keywords:** layered coordination polymer; oxidative desulfurization; denitrogenation extraction; hydrogen peroxide; lanthanides

## 1. Introduction

One of the main aims of Green chemistry is to minimize the negative impact of the petroleum and chemical industries on the environment and human health. The major sources of air pollution in urban areas are the road fuels [1,2], releasing to the atmosphere different pollutants such as carbon monoxide, ammonia, sulfur dioxide (SO$_2$), nitrogen oxides (NO$_x$) and particulate matter. SO$_2$ and NO$_x$ emissions can have adverse effects for the environment and for human health [3], with harsh regulations being implemented for sulfur content in road fuels to decrease SO$_2$ emissions [4–6]. In the European Union, strict policies implemented the limit to sulfur level in diesel from 2000 ppm in 1993 to 10 ppm presently. On the other hand, the nitrogen containing levels in fuels are not regulated. NO$_x$ emissions results from either the oxidation of nitrogen present compounds in fuels or the oxidation of atmospheric nitrogen at high temperatures [7]. These emissions have already been restrained: the limit for diesel powered light duty vehicles decreased from 0.18 g km$^{-1}$ for the Euro V standard to 0.08 g km$^{-1}$ for Euro VI [8,9].

The industrial process is able to efficiently remove the heterocyclic sulfur and nitrogen compounds (such as benzothiophenes, dibenzothiophenes, thiophenes, quinolines, indoles, carbazoles, acridines, and pyridines, Figure S1 in Electronic Supporting Information) from fuels by hydrotreating processes which require severe conditions (high temperatures >350 °C and high pressure 20–130 atm $H_2$) [4,10]. The presence of nitrogen compounds in fuels even in low concentrations (<100 ppm) affects the efficacy of hydrodesulfurization (HDS) reactions since these compounds are strong inhibitors and promoting catalytic deactivation, caused by their competition with sulfur compounds for the active sites of the hydrotreating catalysts [10–13]. Novel technologies capable of complementing or replacing the industrial HDS and hydrodenitrogenation (HDN) processes are needed in order to obtain ultra-low sulfur and nitrogen levels in fuels by more attractive cost-effective methods.

In the literature, various examples dedicated to the elimination of sulfur compounds from the fuels are described. The most promising processes are the oxidative desulfurization, combining with the liquid-liquid extraction [4–6]. Most of these processes use hydrogen peroxide as the oxidant, because of the high active oxygen content and with water being the sole by-product.

Solid catalysts that allow an easy separation from the reaction media and easy recyclability in consecutive reaction cycles are required. Metal-Organic Frameworks (MOFs) and Coordination Polymers (CPs) comprising organic bridging ligands and metallic centers emerged as promising materials because of their considerable structural diversity and unique properties, such as high porosity, large surface areas and in certain cases high thermal, chemical and hydrolytic stabilities. The investigation and use of these materials as heterogeneous catalysts for desulfurization processes is, however, relatively scarce. Composite materials based on MOFs/CPs have been more regularly applied, employed as host materials to support active homogenous catalysts in their pores or matrices [14–17]. Taking advantage of the possibility of some synthetic modification strategies, the catalytic activity of these materials can be greatly improved [18]. In the last decade, we have been focusing on the design of networks based on polyphosphonic acid ligands and rare-earth metal cations which typically induce the formation of highly robust dense networks [19]. The crystalline hybrid materials designed and developed by our groups typically have a high concentration of acid protons and solvent (typically water) molecules [20]. We are looking to take advantage of this particular structural feature to design better-performing compounds that take advantage of proton mobility and/or exchange. These acid properties can be an important advantage for oxidative desulfurization.

In the present work, a positively charged lamellar coordination polymer based on a flexible triphosphonic acid linker, [Gd($H_4$nmp)($H_2$O)$_2$]Cl$_2$ $H_2$O (**1**) [$H_6$nmp stands for nitrilo(trimethylphosphonic) acid], was used as an acid heterogeneous catalyst in oxidative desulfurization and denitrogenation. Environmentally-friendly conditions (low temperature and $H_2O_2$/S molar ratio, and an ionic liquid as solvent [BMIM]PF$_6$) were employed. The stability and the recycle capacity of the catalyst was also investigated.

## 2. Results and Discussion

### 2.1. Preparation of the Catalyst

Our research group have shown the potential of MOFs and CPs as heterogeneous catalysts in various reactions with great industrial interest. The materials tested were based on polyphosphonic acid, self-assembled with various lanthanide cations. The high number of phosphonate groups proved to be important for the catalytic process: the local large concentrations of acidic protons and solvent molecules allowed high conversions rates with low reaction times for reactions such as sulfoxidation of thioanisole [21], conversion of styrene oxide [22] and conversion of benzaldehyde into (dimethoxymethyl)benzene [23]. We further explore the catalytic activity of a positively charged layered CP obtained by combination of nitrilo(trimethylphosphonic) acid ($H_6$nmp) and Gd$^{3+}$ ions (Figure 1), [Gd($H_4$nmp)($H_2$O)$_2$]Cl·2$H_2$O (**1**). When prepared using a simple one-pot method, this material showed a high catalytic activity in four different organic reactions: alcoholysis of styrene oxide, acetalization of

benzaldehyde and cyclohexanaldehyde and ketalization of cyclohexanone, with conversion above 95% after 1–4 h of reaction [24]. In this work we further explore the catalytic activity of this charged 2D material (charge balanced by chloride ions), in the desulfurization and denitrogenation of a multicomponent model diesel. Compound **1** was prepared using three different methods: one-pot (as reported previously), microwave-assisted and hydrothermal syntheses (Figure 1). All materials present the same crystalline phase with slight differences in crystal morphologies and sizes (Figure S2 in ESI). As previously reported, the use of hydrochloric acid is crucial for the preparation of **1**: the acid allows the protonation of the organic linker, retarding the coordination process, leading to the formation of a more crystalline material. Not only that, the acid is the source of chloride anions that are present in the interlayer spaces, being responsible for the CP charge balancing (we note that the use of $GdCl_3$ as the metallic source does not originate compound **1**). For additional structural details on this compound we refer the reader to our past publication [24].

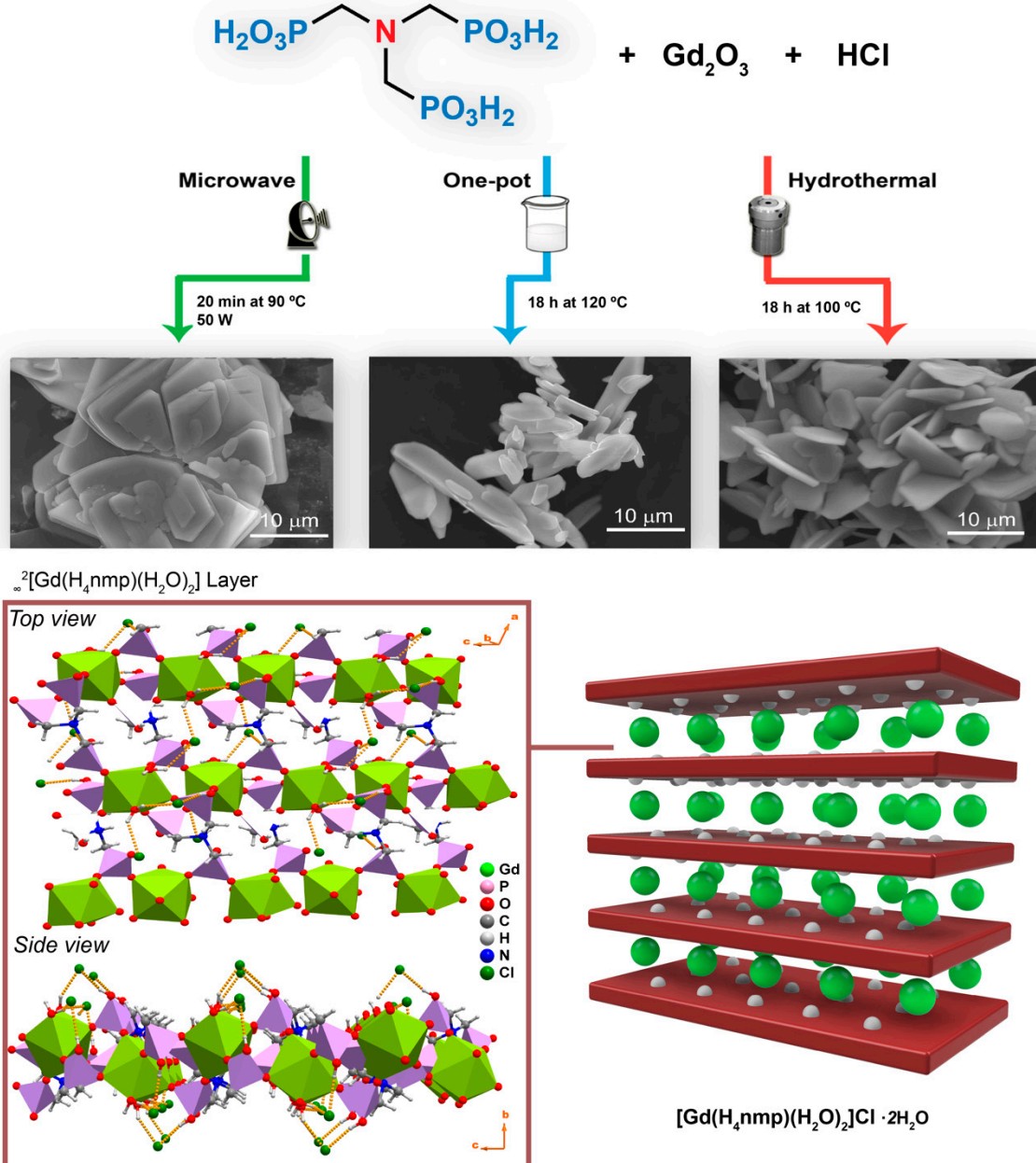

**Figure 1.** Schematic representation of the synthesis and structural features of $[Gd(H_4nmp)(H_2O)_2]Cl \cdot 2H_2O$ (**1**), with emphasis in the chloride anions present in the interlayer spaces.

## 2.2. Optimization of Desulfurization Process

[Gd($H_4$nmp)($H_2O$)$_2$]Cl·2$H_2O$ (**1**) was used as catalyst in the oxidative desulfurization (ODS) of a model diesel composed by the most refractory sulfur compounds toward HDS process in liquid fuels, namely 1-benzothiophene (1-BT), dibenzothiophene (DBT), 4-methyldibenzothiophene (4-MDBT) and 4,6-dimethyldibenzothiophene (4,6-DMDBT) in *n*-octane with a total sulfur concentration of approximately 500 ppm (0.0156 mol dm$^{-3}$) for each compound (for more detailed information of the experimental conditions the reader is directed to the ESI).

Initially catalyst **1**, prepared by the three different methods, was tested and their ECODS profiles are presented in Figure 2. The ionic liquid 1-butyl-3- methylimidazolium hexafluorophosphate ([BMIM]PF$_6$) was used as extraction solvent. Similar desulfurization efficiency was obtained following different preparation procedures. An initial extraction desulfurization of approximately 40% was achieved after 10 min at 70 °C. After the addition of the oxidant the desulfurization increased more appreciably after 40 min and complete desulfurization was found near 2 h. Therefore, the method of catalyst preparation does not seem to have a significant effect in the catalytic efficiency. A small difference is observed between the 40 and 70 min mark and it can be attributed to the overall average particle size. While all methods originated in crystals with plate-like morphology, small differences in size were indeed observed. Contrary to that observed for other materials, the hydrothermal synthesis allows the formation of regular plates with the average size of *ca.* 10 m. On the other hand, microwave synthesis originated plates as agglomerates ranging from 15 to 30 m. Crystals obtained by the one-pot method presented a more irregular crystal morphology, with sizes varying between 5 and 15 m. Nonetheless, the overall small difference of particle size does not seem to have a direct influence in the catalyst activity itself, achieving the same desulfurization efficiency after 2 h of reaction. Their similar crystallinity may indicate that the active catalytic centers in the various catalytic samples are identical.

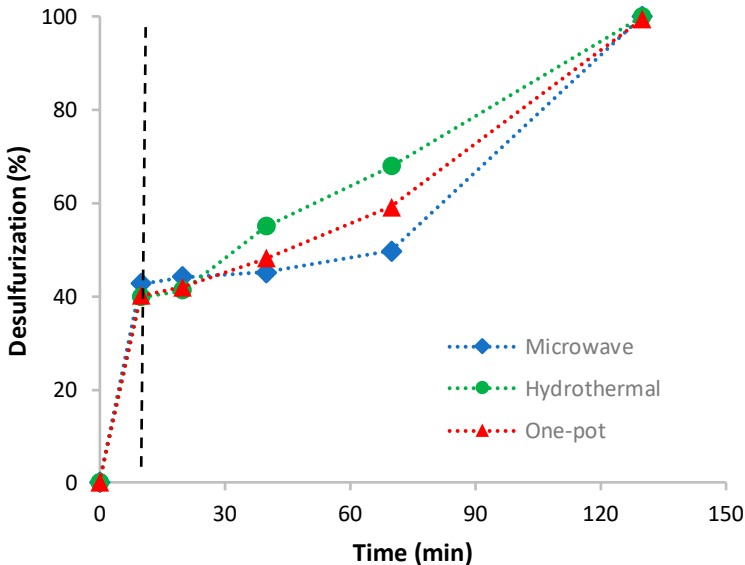

**Figure 2.** Desulfurization of a multicomponent model diesel (2350 ppm S) catalyzed by [Gd($H_4$nmp)($H_2O$)$_2$]Cl·2$H_2O$ (**1**) (20 mg) prepared by different methods (microwave, hydrothermal and one-pot), using equal volumes of model diesel and [BMIM]PF$_6$ as extraction solvent and $H_2O_2$ (0.64 mmol, 30% aq.) at 70 °C. The vertical dashed line indicates the instant that oxidative catalytic reaction was started by addition of oxidant.

Because the catalytic activity was virtually the same for **1** obtained by the various synthetic methods, the following optimization of the ECODS system was performed using [Gd($H_4$nmp)($H_2O$)$_2$]Cl·2$H_2O$ prepared by the one-pot approach (**1op**). Various parameters were investigated, such as extraction solvent, temperature, and oxidant amount, in order to improve the catalytic performance,

the sustainability and cost-efficiency of the process. Three different extraction solvents were investigated: two different ionic liquids (ILs), [BMIM]PF$_6$ and [BMIM]BF$_4$, and the polar organic acetonitrile (MeCN). Figure 3 displays the desulfurization profiles using the various extraction solvents (1:1 model diesel/extraction solvent) and also when no solvent is used. The ECODS system model diesel/[BMIM]PF$_6$ was the most efficient, achieving complete desulfurization after 2 h of reaction. In the absence of extraction solvent, practically no oxidative desulfurization occurred, what indicates an inefficiency of this layered catalyst in the model diesel phase. Using acetonitrile, an initial extraction of 48% of desulfurization was achieved after 10 min. This desulfurization was not, however, increased during the oxidative desulfurization stage, indicating that the catalyst is not active using this solvent extraction. When the solvent extraction used was the IL [BMIM]BF$_4$, a lower initial extraction was achieved (22%) and the desulfurization increased after the addition of the oxidant during the first 30 min to 43%, stabilizing this result after this time. Therefore, the combination of the catalyst **1op** with the [BMIM]PF$_6$ solvent promoted the highest catalytic efficiency, achieving complete desulfurization after only 2 h. ECODS process performed with the 1:1 model diesel/[BMIM]PF$_6$ system without catalyst did not resulted any oxidative desulfurization, i.e, after the initial extraction the desulfurization did not increased after oxidant addition. This result also indicates that the absorptive capacity of the material is negligible. The outstanding catalytic result achieved with the solvent [BMIM]PF$_6$ can be explained by its immiscibility with the oxidant, creating a three phases system (diesel/H$_2$O$_2$/[BMIM]PF$_6$). This prevents direct contact between the catalyst and the oxidant, avoiding a possible catalyst deactivation.

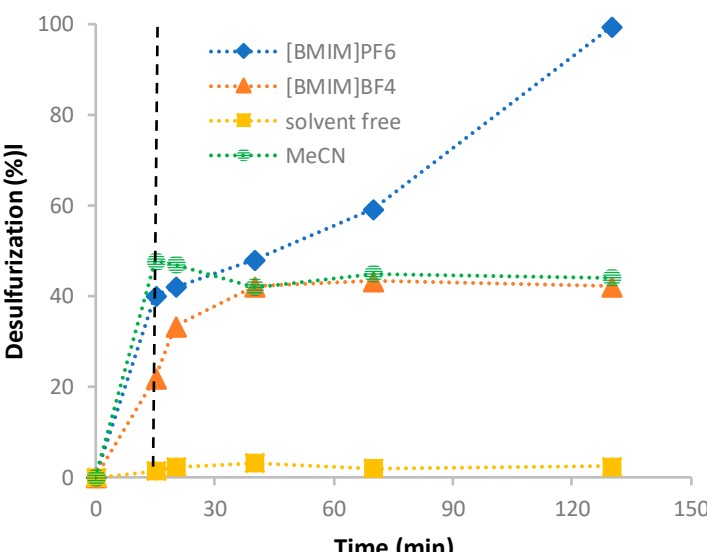

**Figure 3.** Desulfurization of a multicomponent model diesel (2350 ppm S) catalyzed by [Gd(H$_4$nmp)(H$_2$O)$_2$]Cl·2H$_2$O (**1op**) (20 mg), using different extraction solvents (MeCN, [BMIM]PF$_6$/[BMIM]BF$_4$) or absence of this solvent, H$_2$O$_2$ (0.64 mmol, 30% aq.) as oxidant at 70 °C. The vertical dashed line indicates the instant that oxidative catalytic reaction was started by addition of oxidant.

Comparing the oxidative desulfurization of each sulfur-based compound, it was possible to notice that after the initial extraction, 1-BT and DBT are more easily transferred from model diesel to the extraction solvent. As 1-BT possesses the smallest molecular size, this transfer is easier, while the presence of methyl substituents at the sterically hindered positions in 4-MDBT and 4,6-DMDBT makes the extraction more challenging [25–27]. During the oxidative catalytic stage, the 1-BT is the most difficult to be oxidized because after 1 h its desulfurization was 67% in contrast to the 92%, 88% and 80% of total desulfurization for DBT, 4-MDBT and 4,6-DMDBT, respectively (Figure 4). The lower electron density of 1-BT compared with the other studied compounds may explain its lower reactivity [27–29]. The studied dibenzothiophene derivative exhibit similar electron densities on the sulfur atom and

their distinct desulfurization performance is probably caused the steric hindrance promoted by the methyl groups.

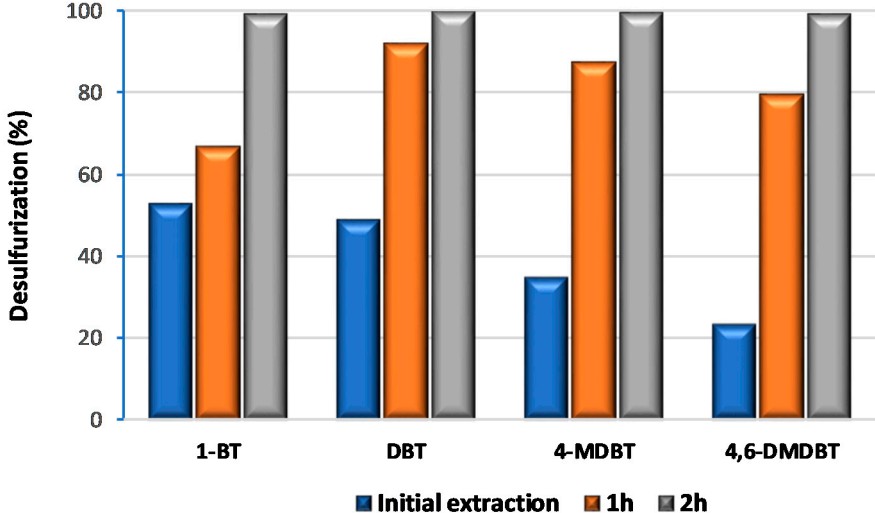

**Figure 4.** Desulfurization results obtained for each sulfur compounds in the model diesel (2350 ppm S) catalyzed by [Gd(H$_4$nmp)(H$_2$O)$_2$]Cl·2H$_2$O (**1op**) (20 mg), using [BMIM]PF$_6$, as extraction solvent, H$_2$O$_2$ (0.64 mmol, 30% aq.) as oxidant at 70 °C.

The effect of temperature (50, 70 and 80 °C) on the oxidative desulfurization process was analyzed and results are summarized in Figure 5. Higher temperatures were not investigated because decomposition of H$_2$O$_2$ could become significant above 80 °C. The increase in the reaction temperature from 50 to 70 °C led to an improvement in the desulfurization rate and resulted in a sulfur-free model diesel product after 2 h of reaction. The desulfurization profile of ECODS at 80 °C demonstrated a rapid increase of desulfurization after the first 30 min of reaction, and after 1 h of reaction 95% of total desulfurization was achieved. Complete desulfurization was, however, only found after 2 h. Therefore, the optimized temperature is 70 °C because complete desulfurization is achieved after 2 h.

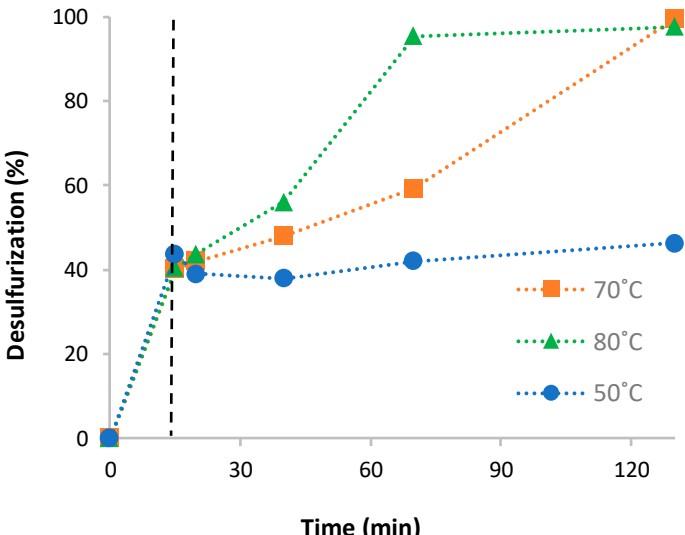

**Figure 5.** Desulfurization of a multicomponent model diesel (2350 ppm S) catalyzed by [Gd(H$_4$nmp)(H$_2$O)$_2$]Cl·2H$_2$O (**1op**) (20 mg), using 0.64 mmol of H$_2$O$_2$ oxidant, [BMIM]PF$_6$ as extraction solvent, at different temperatures (50, 70, 80 °C). The vertical dashed line indicates the instant that oxidative catalytic reaction was started by addition of oxidant.

The oxidant amount is another important factor because it controls, together with the catalyst, the oxidative step through oxygen donation. Three different oxidant amounts were used: 75, 50 and 25 µL, corresponding to 0.64, 0.43 and 0.21 mmol, respectively. Results obtained for the ECODS using [BMIM]PF$_6$ as extraction solvent at 70 °C, are displayed in Figure 6. Using 0.64 and 0.43 mmol of H$_2$O$_2$ a complete desulfurization of the model diesel was obtained after the 2 h. In fact, the desulfurization profile after 1 h of reaction is similar using 0.64 and 0.43 mmol of oxidant. On the other hand, using 0.21 mmol of H$_2$O$_2$ an induction period can be observed until 1 h of oxidation catalytic reaction. This is probably due to the lower interaction of the catalytic active centers with the oxidant caused by its low amount. Therefore, the optimized oxidant amount is 0.43 mmol.

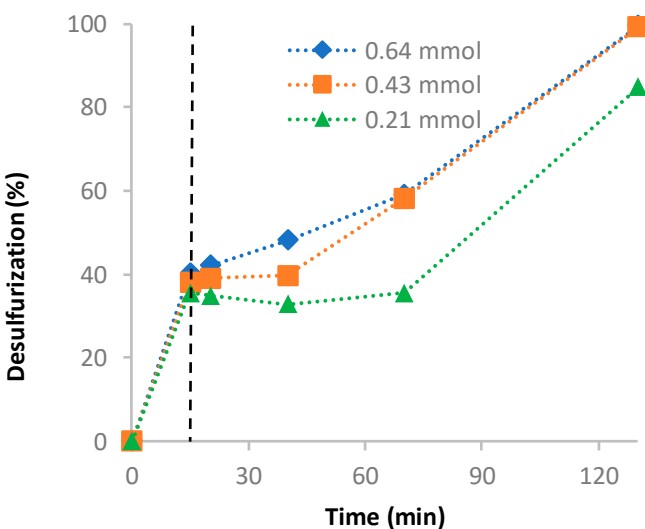

**Figure 6.** Desulfurization of a multicomponent model diesel (2350 ppm S) catalyzed by [Gd(H$_4$nmp)(H$_2$O)$_2$]Cl·2H$_2$O (**1op**) (20 mg), using different amounts of H$_2$O$_2$ oxidant (0.64, 0.43 and 0.21 mmol) and [BMIM]PF$_6$ as extraction solvent, at 70 °C. The vertical dashed line indicates the instant that oxidative catalytic reaction was started by addition of oxidant.

In the next step, the influence of the ratio model diesel/[BMIM]PF$_6$ solvent was studied at 70 °C using 0.43 mmol of oxidant and 20 mg of catalyst. This optimization was performed in order to improve efficiency and sustainability of the ECODS process. Two different ratios were performed: 1:1 and 1:0.5 (Figure S3 in ESI in the Supporting Information). The volume of the extraction solvent phase has an important influence in the initial extraction step because the sulfur compounds are transferred from diesel to this solvent until an equilibrium point is achieved that depends on the nature and amount of this solvent. A higher volume of [BMIM]PF$_6$ may promote a higher initial extraction of non-oxidized sulfur compounds from the model diesel to the polar phase. Figure S3 in ESI shows that the initial extraction using 1:0.5 of model diesel/[BMIM]PF$_6$ was much lower (8%) than when higher volume of extraction solvent was used (1:1 model diesel/[BMIM]PF$_6$, 31%). After the addition of the oxidant (0.43 mmol H$_2$O$_2$), the desulfurization increased much faster when a lower volume of extraction phase was employed. Using the ECODS 1:0.5 model diesel/[BMIM]PF$_6$, 99.5% of desulfurization was found after 1 h, instead of 58% which was achieved using a 1:1 model diesel/[BMIM]PF$_6$ system. The higher catalytic activity and consequent higher desulfurization efficiency obtained in the presence of a lower volume of extraction solvent, must be related to the lower dispersion of the solid catalyst in the ECODS system, which can promote a higher contact with the aqueous oxidant.

## 2.3. Reusability Versus Recyclability

The stability of the layered coordination polymer [Gd(H$_4$nmp)(H$_2$O)$_2$]Cl·2H$_2$O was investigated by its reusability and the recyclability in consecutive ECODS cycles. During all cycles the optimized ECODS conditions were maintained (1:0.5 model diesel/[BMIM]PF$_6$, 0.43 mmol oxidant, at 70 °C).

Two different methods were employed to recover the catalyst after each ECODS cycle. In the "reused" method, the model diesel phase was removed and fresh portions of model diesel and $H_2O_2$ were added without performing any other treatments. In the recycled method the solid catalyst was separated from reaction medium and washed with MeCN and dried. Figure 7 presents the results obtained for three consecutive "reused" and recycled ECODS cycles. The desulfurization efficiency decreases along the consecutive reusing cycles. This behavior has been attributed to the saturation of the extraction phase during the reused process with the oxidized sulfur species, which prevents further transfer of sulfur compounds from the non-polar phase to the IL phase, decreasing the system efficiency [30]. On the other hand, from the recycling method, complete desulfurization was achieved for the three consecutive cycles (Figure 7).

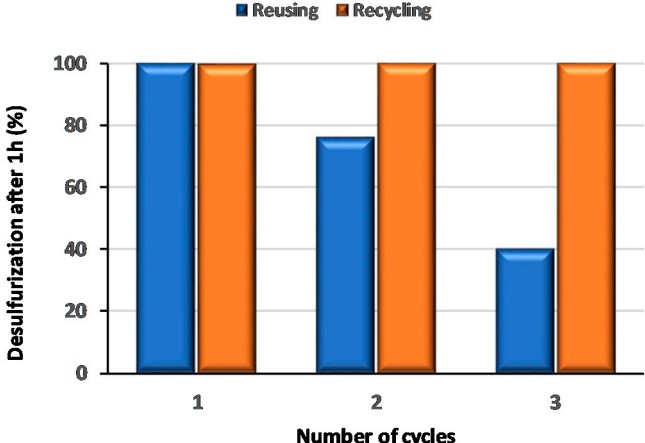

**Figure 7.** Desulfurization results obtained for three consecutive "reused" and recycled ECODS cycles (1 h reaction time), using [Gd(H_4nmp)(H_2O)_2]Cl·2H_2O (**1op**) (20 mg), 0.43 mmol of $H_2O_2$ oxidant and 1:0.5 model diesel/[BMIM]PF_6, at 70 °C.

*2.4. Denitrogenation Process*

Nitrogen compounds present in crude oils are predominantly heterocyclic aromatic compounds. and in smaller amounts as non-heterocyclic nitrogen compounds such as aliphatic amines and nitriles. While the last are easily removed by hydrotreating, heterocyclic nitrogen compounds more difficult to remove. These are classified as basic (mainly pyridine derivatives) and neutral (mainly pyrrole derivatives). Quinoline and indole (Figure S1 in ESI) are representative basic and neutral compounds that are commonly employed as the components of model fuel oils [10]. Using the optimized ECODS conditions, the oxidative denitrogenation efficiency of the layered [Gd(H_4nmp)(H_2O)_2]Cl·2H_2O (1op) catalyst was studied using a model diesel containing indole (200 ppm of N) and quinoline (200 ppm of N). Figure 8 presents the results obtained and it is possible to observe that most of the indole is removed from the model diesel during the first 10 min of initial extraction, instead of 80% achieved by quinoline during this first step of denitrogenation process. After the addition of $H_2O_2$ oxidant, i.e., during the oxidative denitrogenation stage, the removal of quinoline by its oxidation was rapidly increased and after 1 h, with both nitrogen compounds being completely extracted from model diesel. The different behaviors observed for the two nitrogen compounds can be explained by the interaction between the proton/donor molecule indole and the $PF_6^-$ anion, which may promote a selective extraction from the model diesel [31]. Quinoline, containing a six-membered pyridine ring, the electron lone pair on the N atom is not part of the aromatic system and extends in the plane of the ring, being responsible for a negative charge on the N atom, preventing a lower interaction with Lewis acidic IL [10,32–34].

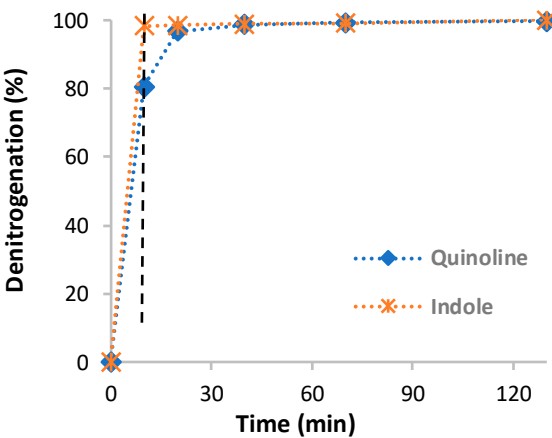

**Figure 8.** Denitrogenation profile for the two refractory nitrogen compounds present in diesel (400 ppm N), catalyzed by [Gd(H$_4$nmp)(H$_2$O)$_2$]Cl·2H$_2$O (**1op**, 20 mg), 0.43 mmol of H$_2$O$_2$ oxidant and 1:0.5 model diesel/[BMIM]PF$_6$, at 70 °C. The vertical dashed line indicates the instant that oxidative catalytic reaction was started by addition of oxidant.

*2.5. Simultaneous Desulfurization and Denitrogenation Processes*

In this work, simultaneous desulfurization and denitrogenation were also performed with a multicomponent model diesel containing the previous studied sulfur compounds (1-BT, DBT, 4-MDBT and 4,6-DMDBT, 2350 ppm of S) and indole (200 ppm of N) and quinoline (200 ppm of N). When the desulfurization and denitrogenation were performed separately, all compounds (nitrogen and sulfur) were extracted under the optimized conditions (20 mg of catalyst, 0.43 mmol of H$_2$O$_2$ and 1:0.5 model diesel/[BMIM]PF$_6$, at 70 °C). However, when denitrogenation and desulfurization were performed simultaneously, practically only nitrogen compounds were extracted from the model diesel to the [BMIM]PF$_6$ phase after the addition of the oxidant (Figure S4 in ESI). Complete removal of N compounds was also achieved after 2 h, instead of 1 h obtained with the single denitrogenation process (Figure 8). These results may be related to the use of an insufficient amount of H$_2$O$_2$, since a competitive oxidation between N and S.

One other parameter that can contribute to the absence of oxidative desulfurization is the low volume of extraction solvent to accommodate both S and N compounds. Therefore, the simultaneous desulfurization and denitrogenation was performed using 0.64 mmol of H$_2$O$_2$ and equal 1:1 of model diesel/[BMIM]PF$_6$ system. An appreciable improvement of the desulfurization profile was found while denitrogenation profile was not strongly affected, using this excess of oxidant and the equal volume of extraction solvent as the diesel phase (Figure 9). Complete denitrogenation was achieved after 2 h (99% after 1 h) and for the same time the desulfurization attained 88% (92% after 3 h). These results indicate that the removal of sulfur is affected by the presence of nitrogen, because complete desulfurization was achieved after 2 h under these experimental conditions using a single sulfur model diesel (Figure 3). This effect is mainly noticed during the oxidative catalytic stage because the initial extraction of each compound was not significantly affected by the presence of S and N in the same model diesel. Figure 10 depicts the extraction for each S and N compound. It is possible to notice that the sulfur compounds more difficult to desulfurize during the oxidative catalytic stage are 1-BT and the 4-DMDBT. Many authors reported that nitrogen compounds are significantly better extracted than sulfur compounds when ILs are used as extraction solvents [35]. Because no oxidized products were detected in the model diesel phase, the oxidative catalytic stage occurs in majority in the [BMIM]PF$_6$ phase and, therefore, the efficiency of desulfurization and denitrogenation processes are strongly dependent of the capacity of S/N extraction.

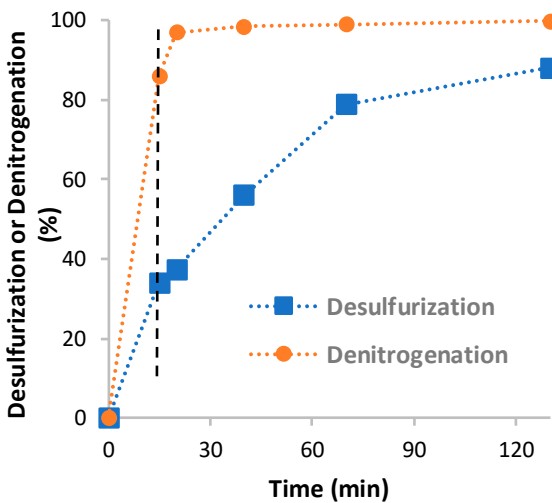

**Figure 9.** Denitrogenation and desulfurization profile of a model diesel containing approximately 400 ppm N and 2200 ppm of S, catalyzed by [Gd(H$_4$nmp)(H$_2$O)$_2$]Cl·2H$_2$O (**1op**) (20 mg), 0.64 mmol of H$_2$O$_2$ oxidant and 1:1 model diesel/[BMIM]PF$_6$, at 70 °C. The vertical dashed line indicates the instant that oxidative catalytic reaction was started by addition of oxidant.

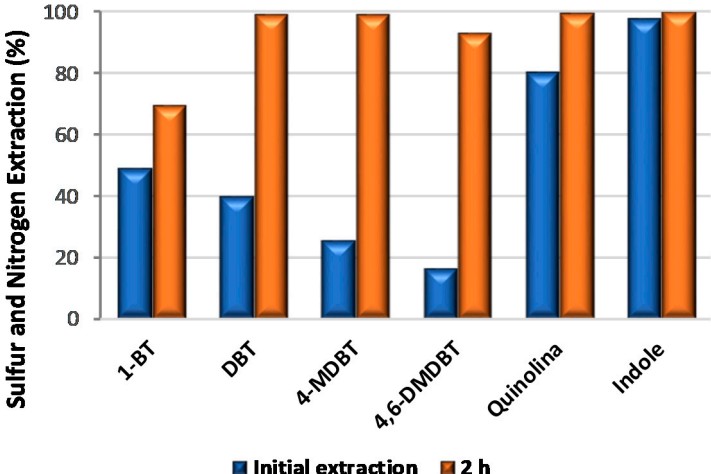

**Figure 10.** Desulfurization results obtained for each S and N compounds in the model diesel catalyzed by [Gd(H$_4$nmp)(H$_2$O)$_2$]Cl·2H$_2$O (**1op**) (20 mg), using [BMIM]PF$_6$, as extraction solvent, H$_2$O$_2$ (0.64 mmol, 30% aq.) as oxidant, at 70 °C.

The catalytic efficiency of [Gd(H$_4$nmp)(H$_2$O)$_2$]Cl·2H$_2$O/[BMIM]PF$_6$ system was studied for various cycles of simultaneous desulfurization and denitrogenation of multicomponent S/N model diesel. This method was chosen instead of the recycling process to face sustainability and cost effectivity that are important topics to consider for industrial application.

## 2.6. Reusability Studies

Reusability experiments were performed using an excess of 0.64 mmol of oxidant and a ratio of 1:1 of model diesel/[BMIM]PF$_6$. Figure 11 displays the results obtained for the three consecutive cycles presenting the data obtained after the initial extraction (before oxidant addition) and after 2 h of oxidative catalytic stage. The initial extraction is not affected by the three reusing cycles. For the denitrogenation process the transfer of N compounds to the extraction phase still increased along the cycles (86, 93, 97% for the 1st, 2nd and 3rd cycles, respectively). The oxidative denitrogenation efficiency is maintained throughout the various cycles and complete denitrogenation was achieved after 2 h. The same is not, however, observed for the oxidative desulfurization, where a decrease

of efficiency was found from the 1st to the 2nd and the 3rd cycles: 88, 62 and 54%, respectively. The efficiency of the desulfurization process is, therefore, diminished by the presence of N compounds during the various consecutive cycles. The higher difficulty of desulfurization process in the presence of N compounds must be even more pronounced during the reusing cycles because the extraction phase became more saturated with N and S compounds from the 1st to the 2nd and to the 3rd cycles. Desulfurization is more sensitive to deactivation during reusing because nitrogen compounds are significantly better extracted than sulfur ones using IL as extraction solvents [35].

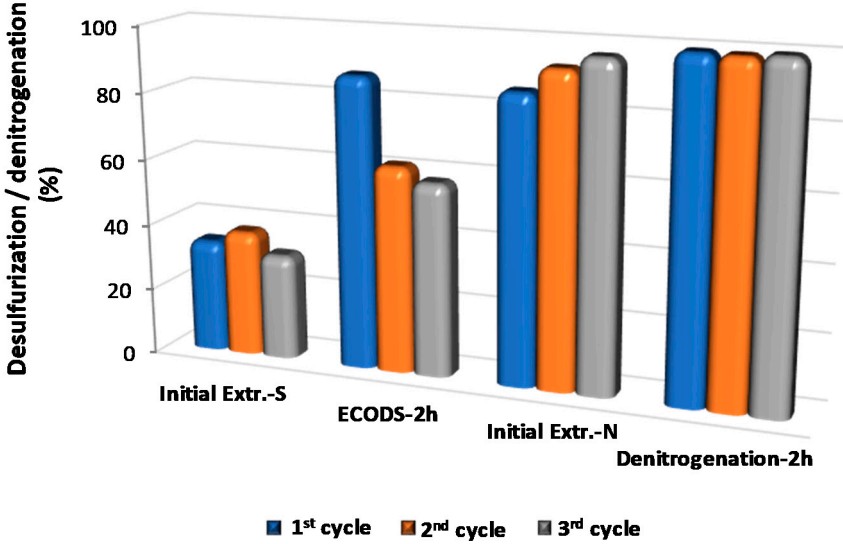

**Figure 11.** Desulfurization and denitrogenation results obtained for three consecutive reused cycles (initial extraction and after 2 h of catalytic oxidation), using [Gd(H$_4$nmp)(H$_2$O)$_2$]Cl·2H$_2$O (**1op**) (20 mg), 0.64 mmol of H$_2$O$_2$ oxidant and 1:1 model diesel/[BMIM]PF$_6$ system, at 70 °C.

*2.7. Structural Stability of [Gd(H$_4$nmp)(H$_2$O)$_2$]Cl·2H$_2$O*

The chemical robustness and structural stability of the catalyst was evaluated by performing several characterizations techniques of the solid after the ODS and ODN processes. As depicted in Figure 12, [Gd(H4nmp)(H$_2$O)$_2$]Cl·2H$_2$O (1) suffers a single-crystal-to-single-crystal transformation under these catalytic conditions. Remarkably, 1 withstands the presence of H$_2$O$_2$. Nevertheless, high temperatures led to the complete transformation of 1, even after the first cycle. [Gd(H4nmp)(H$_2$O)$_2$]Cl·2H$_2$O (1) low stability under harsh conditions was previously reported by us. [36] The catalytic structure **1** suffers a single-crystal-to-single-crystal transformation at high temperatures and relative humidity, resulting in a different 2D layered material. In this case, the combination of high temperature and the presence of the IL led to another of such structural transformation which resulted in yet another distinct, unknown phase.

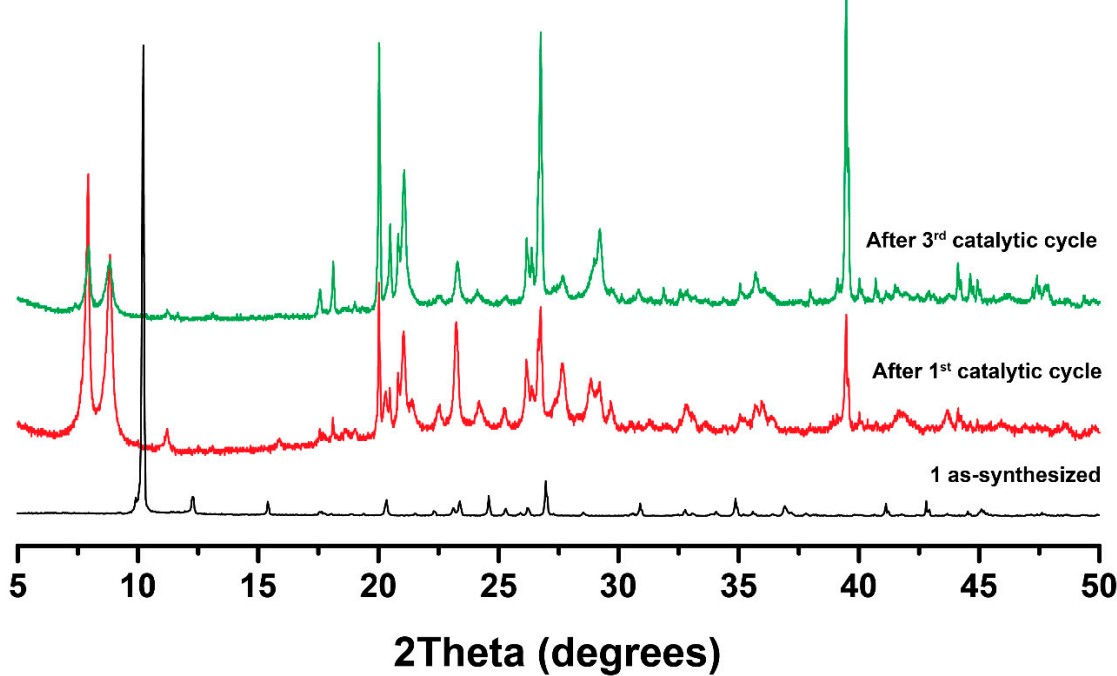

**Figure 12.** Powder X-ray diffraction patterns of the layered [Gd(H$_4$nmp)(H$_2$O)$_2$]Cl·2H$_2$O, before and after the 1st and 3rd ECODS cycles.

Although the structural elucidation of the catalytic material using conventional powder X-ray analysis was not possible, the remaining solid state characterization allowed us to draw some conclusions (see Section 3 in ESI document for further details). The transformation is accompanied with the release of the chloride anion (possibly in the form of hydrochloric acid, very much as recently observed by us). This release is evident by the elemental distribution analysis performed by energy dispersive X-ray spectroscopy (EDS) in the transformed material (Figure S5 in ESI), and supported by previous transformation of this material under different conditions [36]. After the catalytic process, though the particles appear contaminated with sulfur and fluoride species (which can be attributed to residual IL), the quantification of the Gd and P elements could still be performed. This led to a P:Gd ratio of 3:1 after the catalytic use (maintaining the one Gd$^{3+}$ center per each H$_6$nmp linker), which is the same as determined for the initial layered structure. Therefore, we deduce that the observed structural transformation was not accompanied with the loss of either of the basic building units of the compound. In addition, FT-IR analysis (Figure S6 in the Supporting Information) reveals the presence of virtually the same vibrational bands as for **1**, characteristic of phosphonate-based materials, suggesting that no significant changes in coordination modes were observed.

While the crystalline structure could not unveiled, a Pawley refinement was performed (data not shown). The calculated new unit cell (cell parameters: a = 7.91 Å, b = 10.33 Å, c = 11.59 Å; α = 103.95°, β = 95.46° and γ = 90.64°) suggests a change to a less symmetric cell (from *Ia* to possibly *P*-1), with a decrease more accentuated in one of the cell axis (from 17.48 to 10.33 Å). Because 1 is a 2D layered material, the release of the chlorine anions (that are present in the interstitial space acting as charge-balancing anions) might lead to a significant decrease in the interlayer distance, resulting in registered variations of the unit cell parameters. Once again, this structural behavior agrees well with the previously reported transformation of **1** [36,37].

While the powder X-ray diffraction data confirm that the crystalline structure of layered [Gd(H$_4$nmp)(H$_2$O)$_2$]Cl·2H$_2$O is modified after one catalytic cycle, we emphasize that the new structure remained active and without structural change after three consecutive ECODS cycles (Figure 12). The new compound retains the same plate-like crystal morphology.

## 3. Conclusions

The ionic lamellar coordination polymer based on a flexible triphosphonic acid linker, $[Gd(H_4nmp)(H_2O)_2]Cl_2 \cdot H_2O$ ($H_6nmp$ stands for fornitrilo(trimethylphosphonic) acid) presented a high catalytic efficiency to oxidize the most refractory sulfur and nitrogen compounds present in real diesel (mainly dibenzothiophene derivative, indole, and quinolone). The different methods followed for its preparation (microwave, one-pot, hydrothermal) originated some morphological differences, as the size and shape of obtained particles; however, this did not influence its catalytic performance. Using the model diesel/$H_2O_2$/[BMIM]$PF_6$ system, complete desulfurization and denitrogenation were achieved after 2 h of reaction using sulfur or nitrogen model diesel, respectively. The ionic liquid [BMIM]$PF_6$ was the extraction solvent selected since its efficiency was higher than MeCN and other more hydrophilic ionic liquids. When the single model diesel was replaced by the multicomponent S/N model diesel, the desulfurization efficiency decreases from 100% to 88% after 2 h, while the denitrogenation effectivity was maintained. The initial extraction (before oxidant addition) for sulfur and nitrogen was maintained when single model diesels were replaced by the multicomponent diesel. However, the extraction of nitrogen compounds is higher (86%) than the sulfur compounds (36%), what contribute largely for the higher efficiency of denitrogenation process. The recycle capacity of the lamellar catalyst was studied for consecutive desulfurization processes and the catalytic efficiency was maintained between cycles. This result indicates that $[Gd(H_4nmp)(H_2O)_2]Cl_2 \cdot H_2O$ is a stable catalyst, although some structural adjustment occurred to form the active heterogeneous catalyst. An improvement in the reuse capacity of the diesel/$H_2O_2$/[BMIM]$PF_6$ system need to be performed in the near future, since the desulfurization process catalyzed by the lamellar material loss efficiency in consecutive ECODS cycles, probably caused by the saturation of the extraction [BMIM]$PF_6$ phase with sulfur and nitrogen compounds.

**Supplementary Materials:** The following are available online at http://www.mdpi.com/2073-4344/10/7/731/s1, Figure S1: The representative sulfur and nitrogen compounds used in this work to prepare model diesels. 1-benzothiophene (1-BT), dibenzothiophene (DBT), 4-methyldibenzothiophene (4-MDBT) and 4,6-dimethyldibenzothiophene (4,6-DMDBT), Figure S2: Powder X-ray diffraction and SEM images of $[Gd(H_4nmp)(H_2O)_2]Cl \cdot 2H_2O$ (1) obtained using different experimental methods (op—one-pot; mw—Microwave Assisted; ht—Hydrothermal), Figure S3: Desulfurization of a multicomponent model diesel (2350 ppm S) catalyzed by layered MOF $[Gd(H_4nmp)(H_2O)_2]Cl \cdot 2H_2O$ (20 mg), using 0.43 mmol of $H_2O_2$ oxidant and different volume of [BMIM]PF6 extraction solvent (1:1 and 1:0.5 model diesel/[BMIM]$PF_6$), at 70 °C. The vertical dashed line indicates the instant that oxidative catalytic reaction was started by addition of oxidant, Figure S4: Denitrogenation and desulfurization profile of a model diesel containing approximately 400 ppm N and 2200 ppm of S, catalyzed by layered MOF $[Gd(H_4nmp)(H_2O)_2]Cl \cdot 2H_2O$ (20 mg), 0.43 mmol of $H_2O_2$ oxidant and 1:0.5 model diesel/[BMIM]PF6, at 70 °C. The vertical dashed line indicates the instant that oxidative catalytic reaction was started by addition of oxidant, Figure S5: SEM, mapping and EDS spectra of compound $[Gd(H_4nmp)(H_2O)_2]Cl \cdot 2H_2O$ (1) after catalytic use for one cycle of ECODS, with a P:Gd ratio of 3:1, Figure S6: FT-IR spectra of layered $[Gd(H_4nmp)(H_2O)_2]Cl \cdot 2H_2O$ before and after catalytic use for one ECODS cycle.

**Author Contributions:** Conceptualization, F.A.A.P and S.S.B.; Data curation, F.M. and R.F.M.; Formal analysis, F.M. and R.F.M.; Funding acquisition, S.S.B.; Investigation, F.M.; Project administration, S.S.B.; Supervision, S.S.B. and F.A.A.P.; Validation, F.A.A.P. and R.F.M.; Visualization, F.M.; Writing—original draft, S.S.B.; Writing—review & editing, F.A.A.P and S.S.B. All authors have read and agreed to the published version of the manuscript.

**Funding:** This work was partly funded through the project REQUIMTE-LAQV (Ref. UID/QUI/50006/2019) and the project GlyGold, PTDC/CTM-CTM/31983/2017, and partially developed within the scope of the project CICECO-Aveiro Institute of Materials, UIDB/50011/2020 & UIDP/50011/2020, LAQV-REQUIMTE (UIDB/50006/2020) and CQE (UIDB/00100/2020) research units, financed by national funds through the FCT/MCTES (Fundação para a Ciência e a Tecnologia / Ministério da Ciência, Tecnologia e Ensino Superior) and when appropriate co-financed by FEDER (Fundo Europeu de Desenvolvimento Regional) under the PT2020 Partnership Agreement. FCT is also gratefully acknowledged for the Junior Research Position CEECIND/00553/2017 (to RFM).

**Conflicts of Interest:** The authors declare no conflict of interest.

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
