# Peer review of "High Catalytic Efficiency of a Layered Coordination Polymer to Remove Simultaneous Sulfur and Nitrogen Compounds from Fuels"

_catalysts, doi:10.3390/catal10070731_

Round 1

Reviewer 1 Report

I have read and reviewed the structure and catalyst reusability of [Gd(H4nmp)(H2O)2]Cl2H2O showing high catalyst performance efficiency for simultaneous SOx and NOx reduction from diesel.   First of all, although the results of catalytic performance were well shown, I would like to add more information on the overall academic considerations, and it is necessary to emphasize the excellence of the catalyst by adding more cahracterization to the catalyst.   If only this part is supplemented, it will be a good paper.

Author Response

Response: The reviewer is correct that additional characterization (especially single crystal X-ray diffraction analysis) would be ideal for the full understanding of the catalytic material. However, all solid-state characterization which was possible to be performed is presented in the manuscript and in the Supporting Information. A more detailed discussion of this characterization was included in the revised manuscript (section 2.6, pages 12-13), particularly some initial unit cell parameters derived from powder X-ray diffraction studies. We emphasize that the compound is notorious to transform itself into other phases by release of the chloride anions present in the interlayer space. This was recently reported by us: see Chem. Sci., 2020, Advance Article - https://doi.org/10.1039/D0SC01762K. In this paper the material transforms itself once again into a completely distinct material whose structure could not be unveiled using standard techniques.

Reviewer 2 Report

The manuscript assigned to catalysts-837574 presents the layered coordination polymer working on desulfurization and denitrogenation of the model diesel simultaneously. The preparation methods for the [Gd(H4nmp)(H2O)2]Cl·2H2O], solvents and amounts of the oxidants were well-optimized in system. The manuscript seems to be possibly published after revisions of the statements. Details are as follow. 

1. It is clear that the prepared [Gd(H4nmp)(H2O)2]Cl·2H2O] were effective to the desulfurization and denitorogenation of the model diesel in the various solvents. However, the effects of the H2O2 seem more dominant since it clearly changed the perfromance uppon addition to the reaction system. It is recommended to include the experiments without addition of H2O2, otherwise with only addition of H2O2 without Gd[Gd(H4nmp)(H2O)2]Cl·2H2O].

2. As mentioned by authors, the crystal structure of the used [Gd(H4nmp)(H2O)2]Cl·2H2O] was totally disintegrated after one catalytic cycle in Figure 12. Please assingn the main crystal diffractions of the used catalysts in the Figure 12. It is ambiguous if the dissolution of the [Gd(H4nmp)(H2O)2]Cl·2H2O] to form acidic hydrocholoric acid and phosphoric acid, which would have acted as desulfurization and denitrogenation agents.

Author Response

  Reviewer 2

The manuscript assigned to catalysts-837574 presents the layered coordination polymer working on desulfurization and denitrogenation of the model diesel simultaneously. The preparation methods for the [Gd(H4nmp)(H2O)2]Cl·2H2O], solvents and amounts of the oxidants were well-optimized in system. The manuscript seems to be possibly published after revisions of the statements. Details are as follow.

1. It is clear that the prepared [Gd(H4nmp)(H2O)2]Cl·2H2O] were effective to the desulfurization and denitorogenation of the model diesel in the various solvents. However, the effects of the H2O2 seem more dominant since it clearly changed the perfromance uppon addition to the reaction system. It is recommended to include the experiments without addition of H2O2, otherwise with only addition of H2O2 without Gd[Gd(H4nmp)(H2O)2]Cl·2H2O].

Authors Response: The authors acknowledge the Referee for this comment. In fact, no structural modification of the gadolinium phosphonate coordination polymer was observed in the presence of only H2O2. We believe the transformation is induced by increase of temperature in the ionic liquid phase because gadolinium phosphonate coordination polymer shows no structural change in the presence of only H2O2. Unfortunately, no recrystallization under different experimental conditions (or even from transformation process) allowed the isolation of crystals suitable for structure elucidation. This information was further clarified in the revised manuscript (section 2.6, pages 12-12).

  1. As mentioned by authors, the crystal structure of the used [Gd(H4nmp)(H2O)2]Cl·2H2O] was totally disintegrated after one catalytic cycle in Figure 12. Please assingn the main crystal diffractions of the used catalysts in the Figure 12. It is ambiguous if the dissolution of the [Gd(H4nmp)(H2O)2]Cl·2H2O] to form acidic hydrocholoric acid and phosphoric acid, which would have acted as desulfurization and denitrogenation agents.

Authors Response: The authors acknowledge the Referee for this important point. A Pawley refinement was performed (data not shown) to calculate new unit cell (cell parameters: a = 7.91 Å, b = 10.33 Å, c = 11.59 Å; a = 103.95º, b = 95.46º and g = 90.64º) suggests a change to a less symmetric cell (from Ia to possibly P-1), with a decrease more accentuated in one of the cell axis (from 17.48 to 10.33 Å). The transformation is accompanied with the release of the chloride anion (possibly in the form of hydrochloric acid, very much as recently observed by us). This information was added to the manuscript in section 2.6, pages 12-13.

Reviewer 3 Report

The paper “High catalytic efficiency of a layered Coordination Polymer for simultaneous desulfurization and denitrogenation of fuels” describes synthesis of gadolinium phosphonate coordination polymer by three methods and its use for catalytic desulfurization and denitrogenation of model diesel fuel.

I would be happy if the authors could address or comment following points:

Is it possible to dig little bit into the recrystallization of the catalyst because the new phase might be the catalytically active species? Is the recrystalization the result of temperature, H2O2 treatment, or the catalytic reaction? If H2O2 is the cause it might explain the longer induction period at lower H2O2 concentrations. The authors could try to do the recrystallization under hydrothermal/solvothermal conditions in order to elucidate the crystal structure of the new phase.

The new phase formed during the reaction has probably larger unit cell. Did the authors check the porosity of it?

Between the recyclation experiments the catalysts were washed with MeCN. Did the authors analyze the solution to identify the product of desulfurization/denitrogenation? This knowledge could rationalize the drop in catalytic activity for reused catalysts.

In Figure S4 should be used the term denitrogenation instead of desnitrification.

Overall, the study is well conducted and the paper is easy to read. For these reasons I recommend the paper for publishing after minor revisions.

Author Response

  Reviewer 3

The paper “High catalytic efficiency of a layered Coordination Polymer for simultaneous desulfurization and denitrogenation of fuels” describes synthesis of gadolinium phosphonate coordination polymer by three methods and its use for catalytic desulfurization and denitrogenation of model diesel fuel.I would be happy if the authors could address or comment following points: 1- Is it possible to dig little bit into the recrystallization of the catalyst because the new phase might be the catalytically active species? Is the recrystalization the result of temperature, H2O2treatment, or the catalytic reaction? If H2O2 is the cause it might explain the longer induction period at lower H2O2 concentrations. The authors could try to do the recrystallization under hydrothermal/solvothermal conditions in order to elucidate the crystal structure of the new phase.

Authors Response: The authors acknowledge the Referee for this important point. In fact, the new MOF phase formed as the result of the catalytic reaction, obtained by single-crystal-to-single-crystal transformation. We believe the transformation is induced by increase of temperature in the ionic liquid because gadolinium phosphonate coordination polymer shows no structural change in the presence of only H2O2. Unfortunately, no recrystallization under different experimental conditions (or even from transformation process) allowed the isolation of crystals suitable for structure elucidation. This information was further clarified in the revised manuscript (section 2.6, pages 12-12, highlighted at green and blue). 

2. The new phase formed during the reaction has probably larger unit cell. Did the authors check the porosity of it?

Authors Response: The authors acknowledge the Referee for this important question. Unfortunately, the porosity analysis was not performed since residual amounts of [BMIM]PF6 were present after catalytic use and could not be removed even after several cycles of catalyst washing.

3. Between the recyclation experiments the catalysts were washed with MeCN. Did the authors analyze the solution to identify the product of desulfurization/denitrogenation? This knowledge could rationalize the drop in catalytic activity for reused catalysts.

Authors Response: The drop of activity was only observed for the desulfurization process and when the reusing method was followed (Figure 7 and 11). In this reusing method the solid catalyst was not removed from the reactor, i.e. the reaction medium (Ionic liquid phase). Therefore, if some leaching occurred during catalytic reaction, the catalytic active centers are not lost between cycles. On the other hand, in the recycling method, the solid catalyst is removed and washed between cycles; however, in this case no loss of catalyst activity was found. The ICP-OES analysis of P and Gd elements after catalytic use demonstrated that the % of these elements were maintained what suggest the absence of leaching. As referred in the manuscript this drop of desulfurization efficiency is not due to the catalyst but caused by a saturation of sulfur compounds in the extraction phase, i.e. the ionic liquid phase.

4. In Figure S4 should be used the term denitrogenation instead of desnitrification.

Authors Response: The authors acknowledge the Referee for this important correction. This was modified in the ESI document.